# The Effects of the Coating and Aging of Biodegradable Polylactic Acid Membranes on In Vitro Primary Human Retinal Pigment Epithelium Cells

**DOI:** 10.3390/biomedicines12050966

**Published:** 2024-04-26

**Authors:** Georgina Faura, Hana Studenovska, David Sekac, Zdenka Ellederova, Goran Petrovski, Lars Eide

**Affiliations:** 1Department of Medical Biochemistry, Institute of Clinical Medicine, Faculty of Medicine, University of Oslo, 0372 Oslo, Norway; 2CIDETEC, Basque Research and Technology Alliance (BRTA), 20014 Donostia-San Sebastián, Spain; 3Institute of Macromolecular Chemistry, Academy of Sciences of the Czech Republic, 162 00 Prague, Czech Republic; studenovska@imc.cas.cz; 4Institute of Animal Physiology and Genetics, Academy of Sciences of the Czech Republic, 277 21 Libechov, Czech Republic; sekac@iapg.cas.cz (D.S.); ellederova@iapg.cas.cz (Z.E.); 5Department of Cell Biology, Faculty of Science, Charles University, 128 00 Prague, Czech Republic; 6Center for Eye Research and Innovative Diagnostics, Department of Ophthalmology, Oslo University Hospital and Institute for Clinical Medicine, University of Oslo, 0424 Oslo, Norway; goran.petrovski@medisin.uio.no; 7Norwegian Center for Stem Cell Research, Oslo University Hospital, 0424 Oslo, Norway; 8Department of Ophthalmology, University Hospital Centre, University of Split School of Medicine, 21000 Split, Croatia; 9UKLO Network, University St. Kliment Ohridski, 7000 Bitola, North Macedonia; 10Department of Medical Biochemistry, Oslo University Hospital, 0424 Oslo, Norway

**Keywords:** RPE, nanofibrous membrane, AMD, eye, DNA damage, gene expression, retina, retinal pigment epithelium

## Abstract

Age-related macular degeneration (AMD) is the most frequent cause of blindness in developed countries. The replacement of dysfunctional human retinal pigment epithelium (hRPE) cells by the transplantation of in vitro-cultivated hRPE cells to the affected area emerges as a feasible strategy for regenerative therapy. Synthetic biomimetic membranes arise as powerful hRPE cell carriers, but as biodegradability is a requirement, it also poses a challenge due to its limited durability. hRPE cells exhibit several characteristics that putatively respond to the type of membrane carrier, and they can be used as biomarkers to evaluate and further optimize such membranes. Here, we analyze the pigmentation, transepithelial resistance, genome integrity, and maturation markers of hRPE cells plated on commercial polycarbonate (PC) versus in-house electrospun polylactide-based (PLA) membranes, both enabling separate apical/basolateral compartments. Our results show that PLA is superior to PC-based membranes for the cultivation of hRPEs, and the BEST1/RPE65 maturation markers emerge as the best biomarkers for addressing the quality of hRPE cultivated in vitro. The stability of the cultures was observed to be affected by PLA aging, which is an effect that could be partially palliated by the coating of the PLA membranes.

## 1. Introduction

Age-related macular degeneration (AMD) is a complex disease that involves the degeneration of the retinal pigment epithelium (RPE), and it leads to dysfunctional photoreceptors and the subsequent loss of vision [1,2]. It is a major cause of permanent vision impairment in developed nations, and the treatment options for AMD are currently limited [3,4]. The direct injection of a suspension of RPE cells into the subretinal space has been proposed for the treatment of AMD, but such an approach has led to multi-layering of the cells along with a lack of polarization and incomplete attachment to the Bruch’s membrane, the native support of these cells [5]. Undoubtedly, having sub-optimal RPE cultivations can compromise the proper function of RPEs [6]. One of the most promising treatment options for AMD is the transplantation of RPE cell-seeded porous membranes in the affected zone, which can provide a mature, healthy monolayer of cells containing tight junctions (a critical condition for a good barrier function) on a scaffold mimicking Bruch’s membrane [7,8]. Materials such as polylactic acid (polylactide, PLA) [2] or silk fibroin [7] have risen as great candidates as cell carriers due to their biodegradability, which would ease their gradual degradation while cells replace the scaffold with native material. Several promising transplantation studies have already been performed on rodents and pigs [8,9]. Electrospun PLA-based membranes have especially exhibited excellent properties in comparison to commercial polyester inserts [10]. In general, one of the most important advantages of ultrathin nanofibrous PLA-based membranes in comparison to commercial membranes is that their thickness corresponds very well to the thickness of healthy Bruch’s membrane to which RPE cells adhere in the eye. It was found that the porosity of prepared PLA-based membranes of 72% is nearly four times higher than the porosity of commercial track-etched membranes. Also, native basal infoldings were found only in nanofibrous substrates in comparison to track-etched membranes [9]. Other authors referred to the fact that a low porosity of membranes and pores smaller than 0.4 µm could negatively affect the long-term survival of cultivated cells and could bring subsequent neuroretina degeneration [11]. For that reason, porous ultrathin PLA-based nanofibrous membranes can offer better support for growing cells, allowing for the physiological flow of nutrients and thus preserving the function and anatomy of the neuroretina. In our previous work, it was proven that a porcine primary RPE cell can be successfully cultivated even on a xenofree PLA-based membrane without using a biomimetic coating [10]. However, although PLA-based membranes appear to be a good starting point as carriers, they usually need to be coated with biomimetic components such as laminin or vitronectin [7,8] in order to ensure a better initial adhesion of cells and avoid the possible detachment of the cell layer during mechanical manipulation in subretinal implantation. Matrigel^®^, which is one of the most popular coating materials for cell-culturing membranes [12,13,14,15], is used in our study as a model system for biomimetic coatings. Matrigel has been widely used in many in vitro studies, and has been further used for in vivo studies, e.g., on adipogenesis [16,17]. Several authors have also described the angiogenic process within Matrigel [18,19].

The inherent quality of hRPE cells used in transplantation therapy for AMD is crucial for the success of the treatment. Several factors can be used in a quality assessment of hRPE cells, including pigmentation [20,21], polarity [12], mitochondrial DNA (mtDNA) damage [22,23,24], and gene expression [25,26]. Pigmentation is a crucial characteristic of hRPE cells as it reflects their maturity and functionality [27]. The presence of melanin pigment is important for the absorption of excess light, protecting the retina from UV damage and allowing for the transport of nutrients to photoreceptor cells [28,29,30]. Different techniques, such as histological staining or measuring the melanin content, can be used to evaluate pigmentation in hRPE cells intended for transplantation, while in other works, pigmentation is evaluated in a more observational way by comparing microscope images [21,31]. Another essential aspect of hRPE quality is polarity, which refers to the organization of RPE cells in a monolayer structure with tight junctions. Polarity is crucial for maintaining the integrity and barrier function of the hRPE layer, and it is needed for nutrient transport and waste removal from photoreceptor cells to prevent the pathogenesis of AMD [32,33]. An assessment of hRPE polarity can be performed using immunocytochemistry to visualize the expression and localization of polarity markers, such as ZO-1 and P-cadherin, or physically by measuring transepithelial resistance (*TEER*) [7,13,34,35,36]. Mitochondria play a vital role in hRPE cells, and dysfunction or damage to mtDNA can lead to cellular oxidative stress and impair the viability and function of RPE cells. Mitochondrial DNA (mtDNA) damage is implicated in AMD, and hence, it is a critical factor in assessing the quality of RPE cells for transplantation [22,24,37,38,39,40]. Besides these functional biomarkers, a gene expression analysis helps identify specific markers associated with hRPE cell differentiation, maturation, and function. The expression levels of genes related to hRPE cell characteristics, such as RPE65, Bestrophin-1, or visual cycle proteins like RGR and LRAT, are examples of biomarkers that can contribute to an estimate of the quality and functionality of RPE cells intended for transplantation [20,41,42,43].

Here, we investigated how the prolonged storage of the PLA membrane influenced the maturation of primary human retinal pigment epithelial cells (hRPE). We evaluated Matrigel-uncoated and coated PLA membranes [8,9,10,11] to determine their influence on hRPE morphology, pigmentation, epithelial barrier function, genomic integrity, and the expression of maturity markers.

## 2. Materials and Methods

Figure 1 provides a graphical abstract of the processing of primary hRPE and subsequent downstream analyses.

### 2.1. Cell Isolation, Culture, Passage, and Seeding

The hRPEs used in this work were obtained from adult cadaver donors (age range of 50–85) without any known ocular diseases following the procedure described elsewhere [44]. Briefly, before the isolation of hRPEs, the anterior segment of the eye (corneo-scleral ring), the lens, the vitreous, and the neural retina were removed carefully without damaging the Bruch’s membrane underneath. The cup of the eye was then gently rinsed with 1 mL of DPBS (Gibco, Sigma, New York, NY, USA) and filled with 1 mL of growth medium (Dulbecco’s modified Eagle’s medium and Nutrient Mixture F-12 medium supplemented at a <1:1> ratio with 10% Fetal Bovine Serum (FBS), 100 units/mL of penicillin, 100 μg/mL of streptomycin, and 0.25 µg/mL of amphotericin B (Ab/Am), Gibco, Sigma, NY, USA). The cells were then carefully scraped by means of a bent-end Pasteur glass pipette, collected, and seeded in 12-well plates (polystyrene, Merck, Corning, NY, USA). The cells were incubated at 37 °C in a 5% CO_2_ atmosphere, and the medium was renewed every 2–3 days. The cells had a passage ≤ 2.

Before performing the experiments, the cultures were carefully rinsed with DPBS and the cells were harvested by incubating the cultures on TrypLE^TM^ (Gibco^®^, Thermo Fisher Scientific, Waltham, MA, USA) for 5 min at 37 °C in a 5% CO_2_ atmosphere. The cells were then seeded on either commercially available inserts (Corning Transwell polycarbonate (PC) membrane cell culture inserts, 6.5 mm diameter, 0.4 µm pore) or coated or uncoated electrospun polylactide-based (PLA) membranes (PLA membranes characterized elsewhere, cell culture insert wit 10 mm diameter, PLA membrane with 0.4 µm pore size, porosity of 72%, membrane thickness of 3.7 µm, and fiber diameter of 380 nm) [2,10] at a density of 1.5 × 10^5^−1.8 × 10^5^ cell/cm^3^. For coating, Matrigel^®^ was used at a concentration of 8.7 µg/cm^3^, which was deposited on target PLA membranes, cultured for 1 h at 37 °C, and subsequently rinsed with DPBS. PLA scaffolds were sterilized with 70% ethanol before cell seeding. Both coated and uncoated PLA membranes were kept moist, as drying the membrane would limit cell attachment. In experiments comparing old and freshly prepared PLA membranes, the old ones were 21 months old, while the fresh ones were 1 month old. In both cases, the PLA membranes were stored in a closed plastic box with a desiccator in the freezer at −20 °C. Cells from passages 0 to 3 were used for the experiments.

### 2.2. Morphology and Pigmentation

As mentioned in the introduction, in some papers, pigmentation was evaluated by performing a simple comparison between microscope images. Herein, with the aim of making this comparison more objective, the relative (%) pigmented area in the pictures obtained from the microscope images of the cultures was estimated using ≥4 pictures per well and processing the images with ImageJ. The images of the cultures were obtained using an Olympus CKX53 microscope equipped with an ALPHA1080A HDMI camera.

### 2.3. TEER Measurements

Transepithelial/endothelial electrical resistance (*TEER*) measurements were performed with cell cultures on PC/PLA membranes by means of a voltohmmeter (Millicell ERS-2 Voltohmmeter, Millipore, Sigma) in order to evaluate the epithelial barrier integrity of the cell monolayer on the inserts. The measurements were performed in a serum-free medium on membranes containing cells (*R_t_*) or not containing cells (*R_b_*) as a blank. The *TEER* values were calculated as described in Equation (1), taking into consideration the area of the membrane (*A_m_*) [35,36].
(1)$$TEER\;\left(Ω\cdot {cm}^{2}\right)=\left[R_{t}-R_{b}\right](Ω)\cdot A_{m}({cm}^{2})$$

### 2.4. DNA Damage and Mitochondrial DNA Copy Number Estimation

Total DNA was isolated from the cells that were snap-frozen after each experiment by using a DNeasy blood and tissue DNA isolation kit (Qiagen, Hilden, Germany) following the instructions given by the manufacturer. The obtained material was quantified (NanoDrop analyses) and used to determine mitochondrial DNA (mtDNA) and nuclear DNA (nDNA) damage, following the protocol described in detail elsewhere [45,46], based on a real-time quantitative polymerase chain reaction (RT-qPCR) analysis. Briefly, a qPCR reaction mixture was prepared with or without TaqI restriction enzyme. The 12S and NDUFA9 primers, which are specific for the amplification of the sequence from *MT-RNR1* (mtDNA) and *NDUFA9* (nDNA), respectively, were used (sequence in Appendix A). The PCR program ran on a StepOne™ Real-Time PCR System (Applied Biosystems™) is described in Appendix A. The mtDNA copy number (mtDNA-CN) was calculated as a ratio between 12S and NDUFA9 copies [47,48].

### 2.5. Gene Expression

Total cellular RNA was isolated from cell pellets at the end of the monitoring period using RNeasy Mini Kit (Qiagen, Germany, #74104) and then reverse-transcribed using a High-Capacity cDNA RT Kit (Thermo Fisher Scientific, Waltham, MA, USA, #4368814). After this step, SYBR Green (Applied Biosystems, Foster City, CA, USA #437659) and specific primers (Appendix A) [14,25,49,50] were used for a quantitative real-time PCR according to the manufacturer’s protocol using the CFX96 Touch Real/Time PCR Detection System (Bio-Rad, Hercules, CA, USA). Relative quantification of gene expression was calculated using the equation $2^{-∆Ct}$. GAPDH was used as a housekeeping gene.

F test and Student’s *t*-test were used for statistical analysis using Microsoft Excel’s data analysis plug-in.

## 3. Results and Discussion

### 3.1. Morphology and Pigmentation

The evaluation of the RPE cell’s gross morphology is typically used as a first approach to phenotype the cells and assess their maturity [6,51]. Ideally, in vitro mature RPE cells in confluence are expected to be pigmented and homogeneously distributed or closely packed in a so-called cobblestone-like morphology [52,53]. In the case of adult hRPE cells, these characteristics are more easily observed for donors at very low passages [7]. In order to obtain a preliminary overall idea of the effect of culturing RPE cells on PLA membranes (both coated and uncoated), we compared the morphology and pigmentation of low-passage (0 ≤ *n* ≤ 2) hRPE cell cultures on commercial PC membrane inserts or PLA scaffolds (coated or uncoated).

When passaging the cells while maintaining the same cell/area density from the initial well plates to the PLA membranes, there was a noticeable improvement in the hRPE cell morphology. The cells on a regular well plate presented heterogeneous distribution, pigmentation, size, and morphology, and in some regions, the cultures seemed to show a more elongated shape (Figure 2A), which is commonly associated with epithelial-to-mesenchymal transition (EMT, i.e., dedifferentiation of the cells) [14,54]. The hRPE cells were then transferred to PC and uncoated and coated PLA membranes (same cell/area density), and they were cultured for 21 days (Figure 2B–D). In all cases, the cells showed a better morphology after undergoing passage onto the membranes, with the PLA membranes achieving a better morphology. Both the coated and uncoated PLA membranes showed a much more homogeneous distribution of cells with ab intracellular pigment, size, and shape, with few cell clusters being present, and the cells formed a cobblestone-like pattern.

hRPE cells are usually rich in pigment particles, such as melanin and lipofuscin, with melanin being the most effective absorber of light and melanosomes being active participants in antioxidant processes, by capturing oxygen free radicals [55,56]. Thus, pigmentation in hRPE cells is an important quality to consider.

The degree of pigmentation along with the in vitro cultivation was quantified next. We observed a large degree of variation in the pigmentation from donor to donor. To illustrate such variability, the individual data points were represented in a scatter plot (Figure 3A). Despite the individual variations, cultivation on PLA membranes demonstrated significantly improved pigmentation (Figure 3). Interestingly, the difference seems to be attributed to the cell attachment attributes, as there was no significant change in the pigmentation with time (Figure 3A). Considering the variability of the data with the cultivation time, they were averaged for each supporting membrane to allow for an easy comparative statistical analysis (Figure 3B). The Student’s *t*-test calculations confirm that the pigmentation on the PLA and PC membranes is non-equivalent, presenting average pigmented areas of around 40% and 25%, respectively.

### 3.2. Transepithelial Resistance and Cell Detachment Analysis

The main functions of RPE cells are to maintain the outer blood–retina barrier and to facilitate proper fluid, nutrient, and metabolite flow between the choriocapillaris and neural retina [2,7,31]. Performing *TEER* measurements is a non-destructive and widely accepted technique for quantifying the integrity of the epithelial cell culture barrier [35,36,57]. The *TEER* values for prenatal hRPE can easily go above 200 Ω∙cm^2^, but these values are rarely achieved by adult hRPE cells [8,31]. As shown in Figure 4, the *TEER* values for the hRPE cultures on PC membranes remain essentially constant and below 20 Ω∙cm^2^. On the other hand, the *TEER* values of hRPE on the PLA membranes increased over the observed time of 30 days, achieving values around 70 Ω∙cm^2^.

In view of the large variability of *TEER* data, we also analyzed the maximum *TEER* values (max *TEER*) from each independent experiment (Appendix A). This comparison also supported the fact that PLA membranes yield a better epithelial barrier than PC membranes. Additionally, we also analyzed the time needed to reach the *TEER* time (Appendix A), finding that longer times were needed to reach max *TEER* values. There was no significant difference between hRPE from female and male donors.

As indicated in Figure 2, Figure 3 and Figure 4, the PLA coating in general provided better pigmentation and a better epithelial barrier. Although a coating is usually used to enable proper cultivation onto plastic dishes, we observed the detachment of hRPE cells from the coated PLA membranes. However, such detachment was spotted mainly in uncoated PLA membranes (Figure 5).

It was also observed for cultures on uncoated PLA that the areas free of cells due to the detachment from the extracellular matrix became repopulated over time (Appendix A). However, the cells appeared not to have induced EMT, as they were well pigmented and had a morphology characteristic of mature hRPE cells. Moreover, peripheral partial detachment (PPD) of the hRPE cultures on commercial PC membranes was observed, and its width was measured (Appendix A). In Appendix A, the width of PPD versus *TEER* values at diverse culture timepoints are shown. On day 2, the *TEER* values were still low (tight junctions between cells likely still did not develop), and no PPD was observed. On day 9 of culturing and during the following days, the PPD increased and the *TEER* values decreased accordingly. In Appendix A, where all the occurrences from day 9 are presented, a clear inverse relation between *TEER* and PPD can be observed.

### 3.3. Impact of Supporting Membrane on RPE Gene Programming

The pigmentation and epithelial barrier markers imply that there is a preference for coated PLA for the in vitro cultivation of primary hRPEs. To examine whether these functionalities are linked to gene programming, we investigated the possible role of culture supports on the expression of hPRE-relevant genes. The bestrophin 1 (BEST1) gene is highly and preferentially expressed in hRPEs [58]. Retinal pigment epithelium-specific 65 kDa protein (RPE65) is a key isomerase responsible for converting all-trans-retinyl ester to 11-cis-retinol, a key process of the visual cycle [43]. The transcription factors paired box 6 (*PAX6)* and SRY-box transcription factor 9 (*SOX9)* regulate the maturation of RPE. Tight junction protein 1 (*ZO-1)* is involved in the formation of a proper epithelial barrier. The gene expression analysis showed a significantly increased expression of BEST1 in both PLA membranes compared to the PC inserts. The impact of a distinct membrane on RPE65 expression showed a similar trend to BEST1 without being statistically significant. No apparent effect was found on any of the other markers (Figure 6 and Appendix A). The increased expression of RPE-specific markers rather than maturation markers may indicate that PLA membranes are better for preserving RPE-specific gene expression rather than stimulating RPE differentiation.

### 3.4. DNA Damage and mtDNA Copy Number Analysis

Several studies have demonstrated a correlation between the integrity of DNA, particularly mtDNA, RPE dysfunction, and AMD, [23,38,59,60,61], which makes mtDNA integrity in RPE cells a promising target for quality assessment. Recent studies in the literature suggested that the function of mitochondria in RPEs strongly impacts the metabolome of these cells [62], while several studies supported the higher susceptibility of mtDNA to oxidative stress in comparison to nDNA, making mtDNA damage a suitable biomarker for assessing oxidative stress [40,61,63,64].

Abnormal mtDNA-CN (usually mtDNA-CN depletion) [45,46,62,65] has been associated with oxidative stress, mitochondrial status (e.g., changes in membrane potential), and diseases (including age-related diseases), such as type 2 diabetes mellitus, myopathies, and many cancers and neurodegenerative disorders [47,48,66,67]. Furthermore, mtDNA-CN can be altered by environmental exposures [66], and it is also related to nDNA epigenetic modifications [67]. Consequently, mtDNA-CN is increasingly being used as a biomarker of mitochondrial dysfunction, oxidative stress, and disease [47,48].

To determine genome integrity, we evaluated DNA damage in mtDNA and nDNA using established methods, as well as the mtDNA copy number. No significant differences were found for any of these factors, indicating that the observed effects were not due to altered genome integrity and/or oxidative stress (Figure 7).

### 3.5. Coating Rescues Old PLA Membrane Functionality

The biodegradability of PLA also entails a limited shelf life. In order to assess the impact of storage time on PLA utilization, we analyzed the morphology and gene expression of hRPE plated on a newly synthesized PLA membrane (new; <1 month) and prolonged storage PLA membranes (old; 21 months).

The aging of the PLA membranes negatively affected the hRPE morphology. As seen in Appendix A, the hRPE cells on new PLA membranes presented pigmentation and a typical polygonal shape, although their morphology was not as good as shown in previous sections, as passage 1 cells (instead of passage 0) were used in this case. On the other hand, the hRPE cells plated on old PLA membranes showed few pigmentations and presented an elongated shape characteristic of EMT (Appendix A). The loss of the differentiated hRPE state in old uncoated PLA membranes was also confirmed by a decreased expression of hRPE markers, such membranes (Figure 8).

Surprisingly, the hRPE cell morphology on coated PLA membranes presented similar morphologies for both old and new PLA membranes (Appendix A). The cells maintained their pigmentation and morphology, while no elongated cells (indicative of EMT) could be observed. In comparison to the uncoated PLA membranes, the gene expression analysis of hRPE cultivated on old coated PLA membranes only showed a decreased expression of the regulatory transcription factor *SOX9* (Appendix A). Despite the downregulation of SOX9, there was no corresponding decrease in the expression of its downstream gene, *BEST1* [58].

These findings imply that the negative effect of the prolonged storage of PLA membranes can be reduced by coating. PLA coating likely provides a more beneficial environment for hRPE cells, potentially by alleviating the adverse effects seen with uncoated PLA membranes.

## 4. Conclusions

Biodegradable membranes are of great interest for cell therapy as they allow for the transplantation of pre-cultured mature monolayers of cells without depending on a permanent scaffold, which is gradually degraded and substituted by the extracellular matrix generated by the cells over time. hRPE cells can be cultured directly on electrospun PLA membranes, with the cells showing a typical cobblestone-like morphology, along with the pigmentation of mature hRPE cells (even better than on regular well plates or commercial PC inserts) and a mature state, but with a certain detachment risk. Herein, it has been shown that coating on PLA membranes does not necessarily improve the morphology, pigmentation, or barrier function of hRPE cells (in comparison to uncoated PLA membranes); however, it prevents hRPE extracellular matrix detachment and permits the culturing of hRPE cells on aged PLA membranes.

Our research demonstrates that biodegradable membranes offer a promising platform for cell therapy, enabling the transplantation of pre-cultured mature monolayers of cells without the need for a permanent scaffold. This approach allows for the gradual degradation of the membrane, which is then replaced by the extracellular matrix generated by the cells over time. Our findings demonstrate that hRPE cells can be successfully cultured directly on electrospun PLA membranes, exhibiting typical cobblestone-like morphology, pigmentation characteristic of mature hRPE cells, and a mature state. While coating the PLA membranes did not necessarily improve the cell morphology, pigmentation, or barrier function compared to uncoated membranes, it prevented the detachment of the hRPE from the extracellular matrix and enabled the culturing of hRPE cells on aged membranes. This research sheds light on the importance of considering factors such as coating and aging when developing biodegradable materials for cell culture applications. By understanding how these variables influence cell behavior, researchers can optimize membrane design to enhance cell attachment, function, and longevity. These insights have significant implications for the field of ophthalmology and regenerative medicine, where biodegradable membranes could play a crucial role in supporting cell-based therapies for retinal diseases and other ocular conditions.

Overall, this study contributes to the growing body of knowledge on biodegradable materials for cell therapy and highlights the potential of PLA membranes to be used as platforms for culturing hRPE cells. Further research in this area could lead to the development of advanced biomaterials that support the growth and function of various cell types, ultimately advancing the field of regenerative medicine and tissue engineering.

## Figures and Tables

**Figure 1 biomedicines-12-00966-f001:**
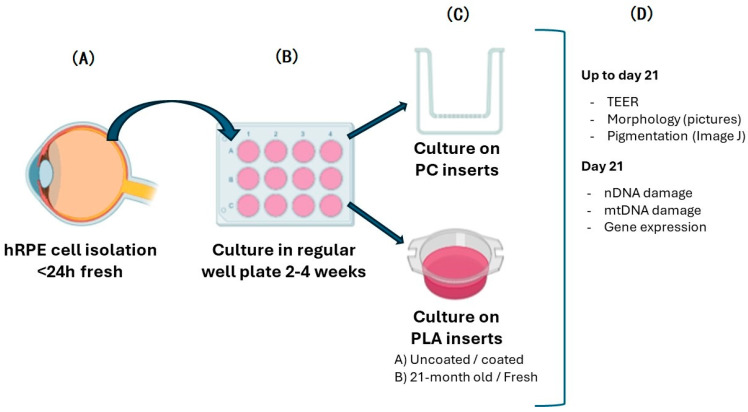
An overview of the experimental design and methods used during this work. (**A**) hRPE cells are isolated from fresh eyes. (**B**) Isolated calls are seeded in regular well plates and typically kept for 2–4 weeks. (**C**) hRPE cells are seeded on diverse types of inserts for comparison. (**D**) For 21 days, the *TEER* values and pictures are taken periodically. On day 21, the cells are collected, and diverse aliquots are used to perform nDNA damage, mtDNA damage, and gene expression tests.

**Figure 2 biomedicines-12-00966-f002:**
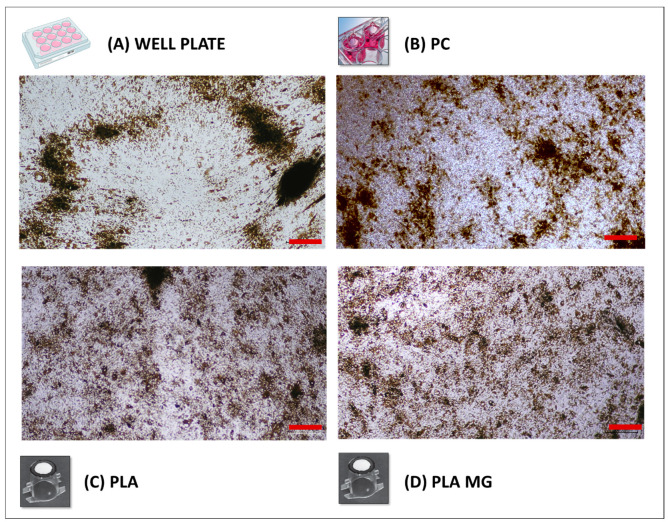
Representative light microscopy images of (**A**) passage 0 hRPE cells on a commercial PC 12-well plate and the same cells seeded at the same cell area density on (**B**) commercial PC, (**C**) uncoated electrospun PLA membranes, and (**D**) coated electrospun PLA membranes after 21 days of culture. Representative images of the membranes without cells are shown in Appendix A for comparison. Red scale bar: 300 µm. PC: polycarbonate; PLA: polylactide; MG: Matrigel coated.

**Figure 3 biomedicines-12-00966-f003:**
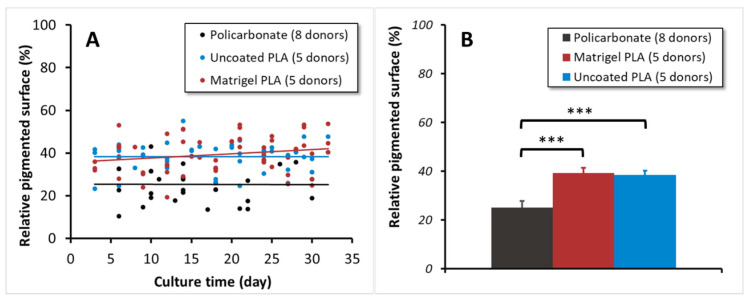
The quantification of pigmentation on the different membranes for the cultivation of hRPE cells. (**A**) A scatter plot showing the relative pigmented surface (RPS) in the images captured for several hRPE cultures on diverse supports specified accordingly. The lines represent the general trend of the data per supporting membrane (Black: Polycarbonate; Blue: Uncoated PLA; Red: Matrigel-coated PLA). (**B**) The mean RPS of the cultures on diverse supporting membranes (PC: polycarbonate; PLA: polylactide). Error bars: confidence interval (95% confidence). *** *p* < 0.001.

**Figure 4 biomedicines-12-00966-f004:**
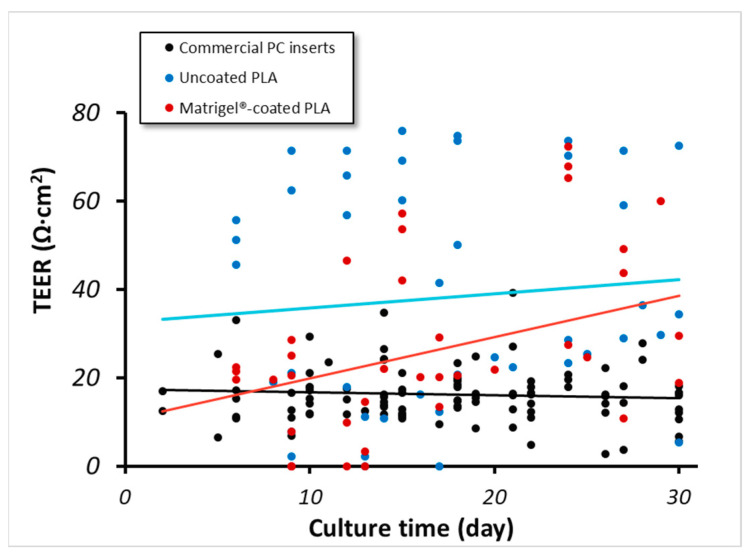
*TEER* values of hRPE cell cultured on commercial inserts (black, 9 ≤ *n* ≤ 22 donors), uncoated PLA (blue dots/line, 2 ≤ *n* ≤ 4 donors), and coated PLA (red dots/line, 2 ≤ *n* ≤ 4 donors). Lines represent general trend of data per supporting membrane (Black: Polycarbonate; Blue: Uncoated PLA; Red: Matrigel-coated PLA). PC: polycarbonate, PLA: polylactic acid.

**Figure 5 biomedicines-12-00966-f005:**
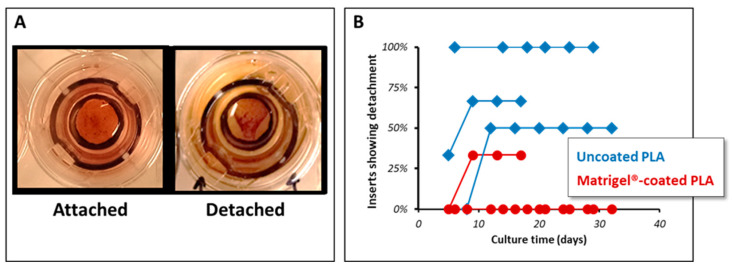
Cell detachment properties of cultivated hRPE cells on different supporting membranes. (**A**) hRPE cell cultures on PLA membranes being well attached (left) and with detaching extracellular matrix (right). (**B**) Representation of amount of PLA membranes showing detachment (merged from 8 different donors in 4 independent experiments).

**Figure 6 biomedicines-12-00966-f006:**
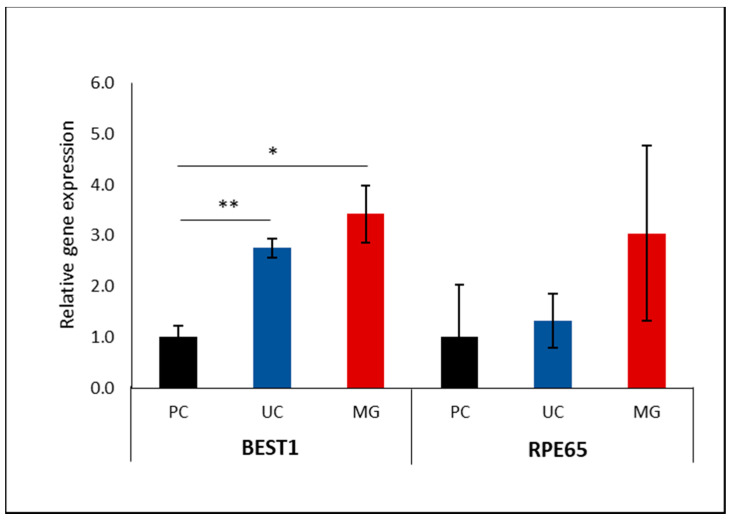
Relative expression of hRPE markers in cultivated hRPSs on different supporting membranes; BEST1 and RPE65 in hRPE cultivated on uncoated and coated PLA membranes in comparison to commercial polycarbonate inserts. Expression of monitored genes in commercial PC inserts was set to 1. Data are shown as ratios. PC: polycarbonate, UC: uncoated PLA membranes, MG: Matrical-coated PLA membranes; 3 donors; error: SEM, * *p* ≤ 0.05, ** *p* ≤ 0.01.

**Figure 7 biomedicines-12-00966-f007:**
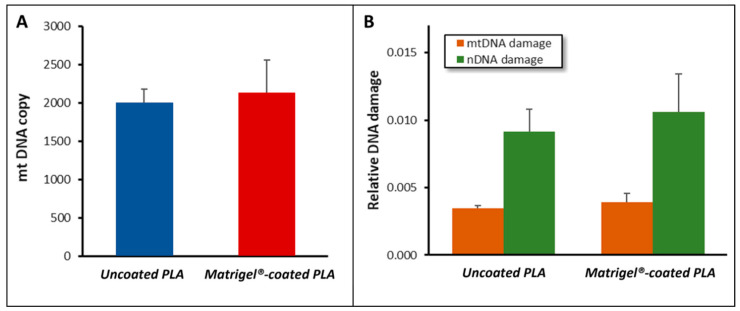
Genome integrity of hRPE cells plated on uncoated versus coated PLA. After 21 days in culture, hRPE cells were collected and genomic DNA was isolated. Cellular mtDNA copy number (**A**) and mtDNA damage and nDNA damage (**B**) in hRPE plated on different membranes were analyzed by qPCR-based methods; 3 donors; error: SEM.

**Figure 8 biomedicines-12-00966-f008:**
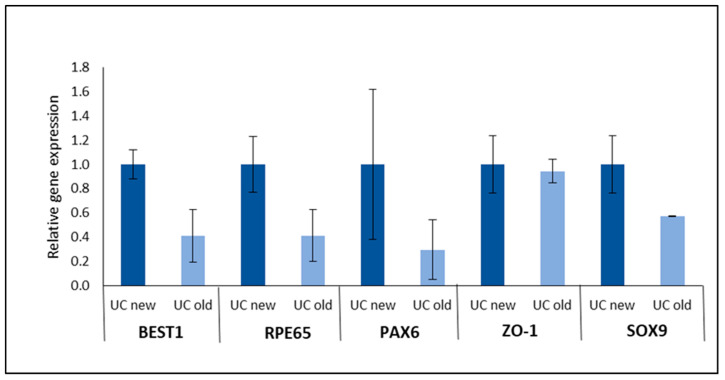
Expression of target genes in hRPE plated on newly synthesized (<1 month) PLA membranes versus 21-month-old uncoated (UC) PLA membranes. Relative gene expressions of *BEST1*, *RPE65*, *PAX6*, *ZO-1*, and *SOX9* were determined by RT-qPCR using *GAPDH* as internal control; 2 donors; error: SEM.

## Data Availability

The data that support the findings of this study are available from the corresponding author upon reasonable request.

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
