# Peer review of "The Effects of the Coating and Aging of Biodegradable Polylactic Acid Membranes on In Vitro Primary Human Retinal Pigment Epithelium Cells"

_biomedicines, 2024, doi:10.3390/biomedicines12050966_

Round 1

Reviewer 1 Report

Comments and Suggestions for Authors

The articles, Effect of the coating and aging of biodegradable PLA membranes on in vitro primary human Retinal pigment epithelium cells, studied the hRPE characteristics such as pigmentation, transepithelial resistance, genome integrity, maturation markers when cells were plated on commercial polycarbonate (PC) versus in-house electrospun polylac-tide-based membranes, both enabling separate apical/basolateral compartments. Their finding shown BEST1/RPE65 as the best biomarkers for addressing the quality of hRPE cultivated in vitro. The work novel and as per scope of journal. My comments to improve the quality of manuscript are as follows.

·         Abstract: Much progress has been done, but still more investigations are needed to  encircle the hRPE parameters associated with successful transplantation, which can be used to further exploit optimized in vitro cultivation conditions. The sentence look very generalized, please improve it.

·         Please cite: 2.2. Morphology and pigmentation, also look for uncited methods if any. In case of novel method, no need of literature citation.

·         Line 83,87 etc: CO2 atmosphere, 2 must be subscript not superscript. Please proof read manuscript carefully during submission,

·         Line 93: For coating, Matrigel®, why mtrigel only? Give statement in manuscript along with this.

·         Line 95: cultured for 1h at 37°C, why this condition, if from literature cite it.

·         Compare the study results with pervious work.

·         What about stability of this cells? 

Author Response

Dear Sir or madam,

Please find our Point-to-point response (in bold) to reviewer no 1’s comments (in italics)

yours sincerely, 
Lars Eide

The articles, Effect of the coating and aging of biodegradable PLA membranes on in vitro primary human Retinal pigment epithelium cells, studied the hRPE characteristics such as pigmentation, transepithelial resistance, genome integrity, maturation markers when cells were plated on commercial polycarbonate (PC) versus in-house electrospun polylac-tide-based membranes, both enabling separate apical/basolateral compartments. Their finding shown BEST1/RPE65 as the best biomarkers for addressing the quality of hRPE cultivated in vitro. The work novel and as per scope of journal. My comments to improve the quality of manuscript are as follows.

  • Abstract: Much progress has been done, but still more investigations are needed to encircle the hRPE parameters associated with successful transplantation, which can be used to further exploit optimized in vitro cultivation conditions. The sentence look very generalized, please improve it.

We have rephrased the sentence and directed it towards the aim of the study (see abstract).

  • Please cite: 2.2. Morphology and pigmentation, also look for uncited methods if any. In case of novel method, no need of literature citation.

We have provided background information in introduction (line 90-93), and additionally materials and methods (line 165, 166).

  • Line 83,87 etc: CO2 atmosphere, 2 must be subscript not superscript. Please proof read manuscript carefully during submission.

We are sorry for overseeing this error. We have updated and proofread manuscript.

  • Line 93: For coating, Matrigel®, why mtrigel only? Give statement in manuscript along with this.

We understand the reviewer’s point. At the same time, our goal was not to identify the optimal coating medium. We provide more information in introduction (line 75-81) to valildate Matrigel as a representative coating agent.

  • Line 95: cultured for 1h at 37°C, why this condition, if from literature cite it.

We did not elaborate on plating time and temperature, but used established practice in the laboratory. The specific condition is in line with available SOPs online. We hope this information is sufficient to address the reviewer’s point.

  • Compare the study results with pervious work.

We refer to the additional input in introduction about quality of hRPE (line 83-108)

  • What about stability of this cells? The RPEs isolated from human cadaver eyes are fully differentiated cells. Their phenotype ex vivo can switch to epithelial-to-mesenchymal transition, which at the early stage of our cultivation should not be the case (EMT is more prevalent in long-term cultures of RPEs). Thus, the stability of the cells in our study should be satisfactory for the experiments performed.

We agree with the reviewer on the point of paying attention to possible EMT. We therefore chose to use low-passage cells (<p2). This information is provided in line 144.

Reviewer 2 Report

Comments and Suggestions for Authors

Study entitled “Effect of the coating and aging of biodegradable PLA membranes on in vitro primary human retinal pigment epithelium cells” by Faura is interesting and design of the paper fine. However, results of the paper seem less convincing as some of the assays showing statistically non-significant results, please see figure 6 and 7 for Genome integrity of hRPE cells and  Expression of target genes in hRPE. This paper might be considered if authors are willing to revise the paper significantly.

In introduction, authors need to elaborate first para and very long compounded statements are given in the introduction section that need to revised substantially.

Page 2, line 68, remove superscript for citation.

Last para of introduction must be revised better clarity of the study.

Have you used permission from concerned authority to use cadaver donors?

In the method section, method used must be cited.

Why excessive inverted comas (“ “) used in the paper?

Rewrite last para of discussion.

Revise conclusion for better clarity.

References are not uniform in the paper, need to check properly.

Comments on the Quality of English Language

Minor edit

Author Response

Dear Sir or madam,
Please find our Point-to-point response (in bold) to reviewer no 2’s comments (in italics)
yours sincerely, 
Lars Eide

Study entitled “Effect of the coating and aging of biodegradable PLA membranes on in vitro primary human retinal pigment epithelium cells” by Faura is interesting and design of the paper fine. However, results of the paper seem less convincing as some of the assays showing statistically non-significant results, please see figure 6 and 7 for Genome integrity of hRPE cells and Expression of target genes in hRPE. This paper might be considered if authors are willing to revise the paper significantly.

In introduction, authors need to elaborate first para and very long compounded statements are given in the introduction section that need to revised substantially.

We thank for the alert, and have worked with improving the sentences as indicated (marked with red text).

Page 2, line 68, remove superscript for citation.

We are sorry for overseeing this format error, it is not fixed.

Last para of introduction must be revised better clarity of the study.

We thank for the alert, and have worked with improving the sentences as indicated.

Have you used permission from concerned authority to use cadaver donors?

We have included a new sentence (Informed Consent Statement, line 444) to specify that approval was provided by consent of next of keen.  The original approval from Regional ethics committee included this requirement, and is now directly visible to readers of the manuscript.

In the method section, method used must be cited.

We hope that the revision related to reviewer 1 is now satisfactorily addressed (pigmentation detection (additions to section 2.1 (line 144) and 2.2 (line 164), and new explanatory figure 1)

Why excessive inverted comas (“ “) used in the paper?

We believe there might be personal preferences, but thank the reviewer for making us aware of the rather frequent use. We have correspondingly replaced all from e.g.  “cobblestone” to cobblestone-like (e.g. line 211, 228, 391), and new / old: line 356-379).

Rewrite last para of discussion.

We have rephrased the last paragraph of the results and discussion to better round off the chapter.

Revise conclusion for better clarity.  

We have improved and expanded the conclusion

References are not uniform in the paper, need to check properly.

We realize that some references were combined, whereas others where not, and have checked for this in the new version where more references are added.

Reviewer 3 Report

Comments and Suggestions for Authors

1.      In this article, the gender of adult cadaver donors is not mentioned. Gender can also affect the obtained results.

2.      For the convenience of the readers, please draw a comprehensive figure to explain the materials and methods.

3.      There is no explanation about the experimental design and the statistical software used in the materials and methods section.

4.      What is the explanation of the “Figure S” that is repeated in different places? It seems to need correction.

Author Response

Dear Sir or madam,
Please find our Point-to-point response (in bold) to reviewer no 3’s comments (in italics)
yours sincerely, 
Lars Eide

In this article, the gender of adult cadaver donors is not mentioned. Gender can also affect the obtained results.

We thank the reviewer for bringing up this point. The TEER values (Figure 4) for hRPE from the two female donors were 21 and 74, while the male donors ranged from 12 to 51). This on uncoated PLA membrane: similar for commercial TW and coated. We have provided the conclusion in line 280 that there was no significant difference between hRPE from female and male donors.

  1. For the convenience of the readers, please draw a comprehensive figure to explain the materials and methods.

We thank for the suggestion to improve the readability, and have included a graphical abstract of the pipeline in materials and methods section (Figure 1; line 117-129).

  1. There is no explanation about the experimental design and the statistical software used in the materials and methods section.

We are sorry for the insufficient description. The new sentence (line 205) now includes the used tools.

  1. What is the explanation of the “Figure S” that is repeated in different places? It seems to need correction.

We believe the reviewer refers to S, which indicates supplementary information. We have included a write-out in the first place (Figure 2 legend; line 233) where it is used.

Reviewer 4 Report

Comments and Suggestions for Authors

In the current manuscript, “Effect of the coating and aging of biodegradable PLA membranes on in vitro primary human retinal pigment epithelium cells,” Georgina Faura and colleagues aimed to study the stability of in-house electrospun polylactide-based membranes compared to commercial polycarbonate and how it influences the maturity of primary human RPEs (hRPE).

The authors analyzed pigmentation, epithelial barrier function, genome integrity, and the expression of maturity markers. They found that the main advantage seems to be that PLA membranes prevent hRPE extracellular matrix detachment and permit the culturing of hRPE cells, indicating that retinal pigment epithelium cells seem to adapt very well to biodegradable PLA membranes.

The authors should emphasize the benefits of these PLA membranes and their further implications.

Author Response

Dear Sir or madam,
Please find our Point-to-point response (in bold) to reviewer no 4’s comment (in italics)
yours sincerely, 
Lars Eide

The authors should emphasize the benefits of these PLA membranes and their further implications.

We have emphasized the benefits of PLA membranes in the conclusion part of the revised version (line 401-420).

Round 2

Reviewer 2 Report

Comments and Suggestions for Authors

Now manuscript is improved. 

Author Response

Dear Sir or Madam,

We interpret the response as the manuscript is sufficiently improved for publication.

Yours sincerely,
Lars Eide

Reviewer 3 Report

Comments and Suggestions for Authors

The revised manuscript is acceptable. Please recheck the order of the figures in the article. It seems that Figure 7 is not cited in the text. 

Author Response

Dear Sir or Madam,

we are sorry for this omitting reference to Figure 7. We have included a reference in text, line 346.

Furthermore and as part of the requested recheck, we have identified additional errors that have been corrected in the final revised version of the manuscript: "Figure 2B" and "Figure 2A" in line 244 and 246 is corrected to "Figure 3" and "Figure 3A", respectively. Lastly, "Figure 2 and 3" (line 282) has been corrected to "Figure 2-4".

We are grateful for this careful review, and are happy to provide an improved version after revision. 

Yours sincerely,

Lars Eide

corresponding author